# Genetics and Genomics of *SOST*: Functional Analysis of Variants and Genomic Regulation in Osteoblasts

**DOI:** 10.3390/ijms22020489

**Published:** 2021-01-06

**Authors:** Núria Martínez-Gil, Neus Roca-Ayats, Mónica Cozar, Natàlia Garcia-Giralt, Diana Ovejero, Xavier Nogués, Daniel Grinberg, Susanna Balcells

**Affiliations:** 1Department of Genetics, Microbiology and Statistics, Faculty of Biology, CIBERER, IBUB, IRSJD, Universitat de Barcelona, 08028 Barcelona, Spain; airun91@gmail.com (N.M.-G.); neroca@clinic.cat (N.R.-A.); monicacozar@ub.edu (M.C.); dgrinberg@ub.edu (D.G.); 2Musculoskeletal Research Group, Centro de Investigación Biomédica en Red en Fragilidad y Envejecimiento Saludable (CIBERFES), ISCIII, IMIM (Hospital del Mar Medical Research Institute), 08003 Barcelona, Spain; ngarcia@imim.es (N.G.-G.); dovejero@imim.es (D.O.); xnogues@parcdesalutmar.cat (X.N.)

**Keywords:** sclerostin, bone, 4C-seq, luciferase reporter assay, HBM

## Abstract

*SOST* encodes the sclerostin protein, which acts as a key extracellular inhibitor of the canonical Wnt pathway in bone, playing a crucial role in skeletal development and bone homeostasis. The objective of this work was to assess the functionality of two variants previously identified (the rare variant rs570754792 and the missense variant p.Val10Ile) and to investigate the physical interactors of the *SOST* proximal promoter region in bone cells. Through a promoter luciferase reporter assay we show that the minor allele of rs570754792, a variant located in the extended TATA box motif, displays a significant decrease in promoter activity. Likewise, through western blot studies of extracellular and intracellular sclerostin, we observe a reduced expression of the p.Val10Ile mutant protein. Finally, using a circular chromosome conformation capture assay (4C-seq) in 3 bone cell types (MSC, hFOB, Saos-2), we have detected physical interactions between the *SOST* proximal promoter and the *ECR5* enhancer, several additional enhancers located between *EVT4* and *MEOX1* and a distant region containing exon 18 of *DHX8*. In conclusion, *SOST* presents functional regulatory and missense variants that affect its expression and displays physical contacts with far reaching genomic sequences, which may play a role in its regulation within bone cells.

## 1. Introduction

Sclerostin (encoded by the *SOST* gene) is a potent antagonist of the canonical Wnt pathway and of bone formation, exerting its function by blocking LRP5/6 co-receptors [1,2]. To perform this function, sclerostin binds to LRP4, which enhances its suppressive effect [3,4,5]. Sclerostin is a monomeric protein with a NxI motif and an eight-membered cysteine knot motif, all of them implicated in LRP5/6 binding [6,7,8].

In humans, *SOST* mutations are associated with inherited high bone mass conditions characterized by excessive bone formation, with a sclerosis phenotype particularly prominent in the skull, jaw and long bones. The clinical spectrum of these conditions ranges from severe craniodiaphyseal dysplasia to non-pathological high bone mass, with the severity of the condition being inversely proportional to sclerostin abundance. Craniodiaphyseal dysplasia (CDD, MIM 122860) is the most severe sclerostin bone dysplasia, characterized by an absence of extracellular sclerostin and a dominant negative effect. It is an extremely rare autosomal dominant condition, with less than 20 cases reported [9,10]. Sclerosteosis (MIM 269500) is another severe sclerostin disease with autosomal recessive inheritance [11,12]. Nine loss-of-function mutations (either truncating or missense) have been described in *SOST* [12,13,14,15,16,17,18,19]. Additionally, three loss-of-function mutations in *LRP4*, have been described in sclerosteosis-2 [4,20]. Van Buchem’s disease (MIM 239100), a third sclerostin clinical entity, is caused by a 52-kb deletion of a regulatory region located 32 kb downstream of *SOST* and necessary for the correct expression of the gene [21,22]. The difference in severity lies in that in patients with Van Buchem, wild-type sclerostin is low, whereas in patients with sclerosteosis it is absent [23,24]. In line with these findings, a heterozygous truncating mutation in *SOST* has been associated with the non-pathological High Bone Mass (HBM) phenotype, which is characterized by unusually dense bones and a very strong skeleton practically eliminating fracture occurrence [25,26,27,28]. Furthermore, it was described that mutations in *LRP5* causing the HBM phenotype showed a loss of affinity of LRP5 for sclerostin [29,30].

Sclerostin is mainly expressed in mature osteocytes, although relatively high levels of mRNA have been found in different tissues [31]. In addition, *SOST*/sclerostin expression is altered in certain pathogenic conditions, including postmenopausal osteoporosis, osteoarthritis and rheumatic joint disease [32]. Genetically modified mice add another line of evidence to the role of *SOST* in bone mineral density (BMD) determination: *Sost*-knock-out (KO) mice show high BMD and bone strength, with a high number of osteoblasts and increased bone formation that reproduce the phenotype observed in patients with sclerosteosis or Van Buchem’s disease [33,34]. In contrast, mice that overexpress this gene present with low bone mass, lower bone resistance, low number of osteoblasts and reduced bone formation [35,36,37,38].

The transcription of *SOST* is finely regulated by a large number of signals. *SOST* expression is controlled by the proximal promoter region (~1.4 kb upstream of *SOST*) and by a 225-bp bone-specific distal enhancer called *ECR5* (Evolutionary conserved region 5), located 32 kb downstream of the gene [31,39]. *ECR5* is deleted in patients with Van Buchem’s disease, and its deletion in mice causes a drastic decrease in sclerostin levels in osteocytes [39]. *ECR5* stimulates the expression of *SOST*, in part through the binding of the transcription factor Myocyte-enhancer factor 2c (Mef2c) [35,39].

Humanized monoclonal antibodies against sclerostin have been developed for the treatment of different bone disorders such as osteoporosis or osteogenesis imperfecta, and they have been approved in Japan, US, EU and Canada [40,41,42,43,44]. Antibody treatment against sclerostin stimulates bone formation, decreases bone resorption and, thus, increases bone mass and strength, and decreases the risk of fragility fracture in both animals and humans [40,41,42,44,45,46].

Taking into account the crucial role of *SOST* in bone growth, we hypothesized that partial loss-of-function mutations in its gene might be associated with variations in BMD in the general population. In this line, we previously identified 2 variants in *SOST* that captured our interest. First, the rare variant rs570754792 located in the extended TATA box, which was only present in 3 women with low bone mass; and second, the low-frequency variant rs17882143 (p.Val10Ile), which modified the signal peptide of sclerostin, and whose minor allele was associated with high BMD [47].

Our objective in this work was to assess the functionality of these variants and to detect the main physical interactors of the *SOST* proximal promoter. We verified the loss of function of rs570754792 and rs17882143, which produced a decrease in gene expression and protein abundance, respectively. In addition, we were able to detect physical interactions of the *SOST* proximal promoter with several enhancers in the region, including the *ECR5* distal enhancer.

## 2. Results

### 2.1. The Minor Allele of a Variant in the Extended TATA Box of SOST Displays Lower Transcriptional Capacity

The rare variant rs570754792 (MAF = 0.004464, gnomAD v2.1.1) lies in the *SOST* proximal promoter, specifically in the extended TATA box. To test its effect, we performed a luciferase reporter assay in Saos-2 cells, testing a 520-bp fragment, which contains either of the variant alleles. Our results show that this region harbors promoter activity (20-fold increased transcription vs. empty vector, for the wild-type allele; Figure 1) and show significant differences between the two alleles tested, where the minor allele displays lower transcription capacity (about 25% less than the major allele; Figure 1). The *COL1A1* promoter (SP construct in [48]), used as positive control, showed 31.6-fold increased promoter activity compared to the empty vector (Figure 1).

### 2.2. Lower Protein Abundance for the Minor Allele of rs17882143, Both in the Intracelullar and Extracellular Spaces

The rs17882143 SNP is a missense change (p.Val10Ile; MAF = 0.0109 gnomAD v.2.1.1) within the signal peptide of the sclerostin protein. To assess whether the change might affect the subcellular localization of the mutated protein or its abundance, we performed western blot studies with extracellular and intracellular protein extracts (Figure 2). In addition to this mutation, we tested 5 of the most frequent sclerostin mutations in the general population according to gnomAD (p.Arg19His, p.Gly150Arg, p.Asn53del, p.Ala28Val, p.Pro111Gln; Figure 2A), as well as two positive controls (p.Val21Leu and p.Cys167Arg; Figure 2A) whose absence in the extracellular space had been previously described [10,17]. All these changes are depicted along the sclerostin protein domains in Figure 2A. We verified that the positive control proteins SOST-p.Val21Leu and SOST-p.Cys167Arg showed a dramatic decrease in the extracellular space. However, while the SOST-p.Cys167Arg showed an increase in the amount of intracellular protein, the SOST-p.Val21Leu showed a decrease in intracellular levels (Figure 2B,C). Similarly to the results of these two positive control proteins (p.Val21Leu and p.Cys167Arg), we observed a lower abundance of SOST-p.Val10Ile and SOST-p.Gly150Arg proteins in the extracellular space and, for the SOST-p.Val10Ile, SOST-p.Val21Ile, SOST-p.Ala28Val, SOST-p.Asn53del and SOST-p.Pro111Gln, a lower abundance in the intracellular compartment. The other mutant proteins behaved similarly to the WT protein (Figure 2B,C).

### 2.3. Physical Interaction between the SOST Proximal Promoter and Several Distal Regulatory Elements in Bone Cells

To identify important regulators of *SOST* expression presence in three bone-related cells (mesenchymal stem cells—MSC-; human fetal osteoblasts—hFOB- and an osteosarcoma cell line—Saos-2-), we performed a 4C-seq assay using the *SOST* proximal promoter (chr17:41838135-41838123, GRCh37/hg19) as a viewpoint (Figure 3). Using an algorithm to discern the significant contacts, we only observed interactions of this viewpoint with sequences included in a genomic region spanning 288 kb centromeric to the viewpoint. Within this region, we confirmed the strong physical interaction between the proximal promoter and the *ECR5* distal enhancer in the 3 cell types. Moreover, we found interactions in the three cell types between the viewpoint and several genomic regions located between *MEOX1* and *ETV4*, described as enhancers according to GeneHancer [49] and displaying strong H3K4me1, H3K4me2, H3K4me3, H3K27ac and H2AFz histone marks in osteoblasts (ENCODE; Appendix A). In addition, we observed an interaction with a region containing exon 18 of the *DHX8* gene that displays strong H3K4me1 and H3K27Ac histone marks in osteoblasts (ENCODE), suggesting a potential regulatory role for this region, too. Finally, we saw a significant interaction with the *DHX8* promoter in hFOB and Saos-2 cells, which displays H3K4me2, H3K4me3, H3K27ac, H3K79m3 and H2AFz histone marks in osteoblasts (ENCODE; Figure 3 and Appendix A).

### 2.4. The rs17882143 (p.Val10Ile) Variant Present in One HBM Woman

To further explore the putative genetic association of *SOST* with bone density phenotypes, we resequenced the *SOST* gene and the *ECR5* enhancer, in 11 HBM women (described in [50]. We identified eight variants in *SOST* and one variant in *ECR5*, all of them previously described (Table 1). Interestingly, we found the variant p.Val10Ile in heterozygosity in one HBM woman (HBM13) with a high sum *Z*-score (lumbar spine + femoral neck BMD *Z*-score = 5.2). In addition, we found the *ECR5* variant rs552004150 in heterozygosity in another HBM woman (HBM16) with a Sum *Z*-score = 5.3. However, it did not co-segregate with the phenotype in the HBM16 family (Appendix A). This variant lies approximately 50 bp from the MEF2C binding site and does not modify the binding of any known transcription factor according to TRANSFAC (Appendix A). Variants rs1237278, rs851058, rs10534024, rs17882143, rs17886183 and rs17881550, found in our HBM cohort, are listed as eQTLs for different genes and tissues in the GTEx database (Appendix A). In addition, variants rs1237278, rs851058 and rs17885799 are described as affecting the binding site of various transcription factors, and the rare variant rs17885799 may generate a new exonic splicing enhancer (ESE) according to SNP function prediction (Appendix A). Furthermore, SNPs rs17883310 and rs17886183 show allele-specific differences in miRNA binding predictions (Appendix A).

## 3. Discussion

Sclerostin is an important bone protein, strongly associated with different bone sclerosing dysplasias. In this work, we undertook a functional study to test two sclerostin variants (rs570754792 and rs17882143) and to define the physical interactions of the *SOST* proximal promoter region. We experimentally demonstrate the partial loss of function of these two variants that cause a loss of expression (rs570754792) and a reduction of protein levels (p.Val10Ile). Furthermore, we detect physical interactions of the *SOST* promoter with *ECR5*, with a region containing exon 18 of the *DHX8* gene and with several enhancers between *MEOX1* and *ETV4*, in 3 bone cell types.

Regarding the rare variant rs570754792, it was found in heterozygosity in only three women of the BARCOS cohort of postmenopausal women of the Barcelona area, all three with BMD values below the mean (lumbar spine BMD: 0.767, 0.694, 0.822 vs. 0.850 g/cm^2^) [47]. According to the established inhibitory role of SOST in the canonical Wnt pathway, we investigated whether *SOST* was expressed at levels higher than normal using a luciferase reporter assay. Surprisingly, we observed just the opposite: lower levels of expression in the presence of the minor allele (T). This result, however, is not totally unexpected, since the rare variant is part of the extended TATA box, and the major and minor alleles represent the canonical and non-canonical base at the position -6, respectively. At this point it is hard to reconcile the low BMD values observed in the three carriers and the lower *SOST* expression levels of the variant. The lack of genomic context of the luciferase experiments might provide an explanation for these contradictory results. On the other hand, it may just be that this variant has low penetrance, and in these women an alternative mechanism is responsible for their phenotype, (i.e., a different mutation or environmental component with a larger effect, masking that of the mutation in the extended TATA box of *SOST*). In fact, it could be a situation similar to that of van Buchem’s disease, in which two copies of a low expressing allele are needed to show the phenotype.

Using western blot analysis, we studied the expression levels of sclerostin containing the SNP rs17882143 (SOST-p.Val10Ile). As this variant is located in the signal peptide of the protein, we expected a lower translocation to the extracellular space. However, we found lower amounts of it, both in the extracellular and the intracellular compartments, which seems to indicate a defect in a previous stage to protein translocation and secretion. This result is supported by an analysis using the SignalIP signal peptide prediction program [51] that predicts no functional differences in the signal peptide. Our result could be explained by reduced transcription or translation levels or increased mRNA or protein degradation. In particular, it has been described that missense mutations can result in an altered secondary mRNA structure which compromises the accessibility of the protein translational machinery [52,53,54]. In our case, this hypothesis is supported by the Mfold web server [55], which predicts a secondary structure that has a different free energy depending on the allele of the SOST-p.Val10Ile. The p.Val10Ile mutation is the most frequent *SOST* missense variant according to gnomAD (MAF 0.0109). In silico functional predictors scored it as benign (SIFT; 0.43), tolerated (Polyphen; 0.001) and likely benign (CADD; 0 and REVEL; 0.123). In a previous study of our group [47], this variant showed nominal association with lumbar spine BMD, where the minor allele (Ile) was protective. Furthermore, we have now found that a woman with an HBM phenotype (with a sum *Z*-score of 5.2) is heterozygous for this variant (Table 1), and it was also detected in 8 additional HBM cases in the work of Gregson et al. [25]. All things considered, it could be speculated that people carrying this mutation show high non-pathogenic BMD due to a reduction of sclerostin expression.

The SOST-p.Gln150Arg protein showed a decreased secretion to the extracellular space compared to the WT. This mutation, which is present in the general population with a frequency of 0.000333, is the third most frequent missense mutation in *SOST* and functional predictors consider it deleterious (SIFT; 0.02), possibly damaging (Polyphen; 0.651) or likely benign (CADD; 23 and REVEL; 0.295). Similarly to SOST-p.Val10Ile, it could be speculated that people carrying p.Gln150Arg may present a non-pathogenic HBM due to low levels of sclerostin in the extracellular space. In the future, BMD analyses of carriers will be helpful to test this prediction.

Regarding the two mutants previously described that we used as positive controls, we obtained similar results to those described by Piters et al., for p.Cys167Arg [17], while we obtained some discrepant results for p.Val21Leu, compared to data from Kim et al. [10]. In particular, we found that intracellular protein abundance of SOST-p.Val21Leu was approximately 50% of that of WT, while Kim et al. [10] showed values similar to wild-type sclerostin. Technical differences, including different cell types (HEK293 vs. Saos-2) and antibodies (anti-myc vs. anti-sclerostin), may explain this discrepancy.

Finally, through 4C-seq we showed for the first time the physical interaction between *ECR5* and *SOST* in 3 bone-relevant cell lines. Further, we identified the interaction of the proximal *SOST* promoter with several putative enhancer elements between *MEOX1* and *EVT4*, which display regulatory histone marks in osteoblasts (ENCODE). Even though they appear as non-tissue specific elements according to GeneHancer, in the future it will be interesting to investigate the role of these enhancers in bone-specific *SOST* expression. We also identified a significant interaction between the promoter and a distal regulatory element overlapping exon 18 of *DHX8*, a gene encoding a spliceosome component, which may be unrelated to bone mass. In the overall emerging picture, several long-distance enhancers may participate in the regulation of *SOST* in bone cells. Whether they work in an additive way or as alternative redundant enhancers is an interesting question for future research.

Sclerostin is a key component of the canonical Wnt pathway in bone and, as such, it plays a crucial role in skeletal development and bone homeostasis. A detailed understanding of its developmental and tissue regulation together with the knowledge of the effect of variation at its locus is key to developing prediction algorithms or therapeutic strategies based on this gene/protein. Our results add functional evidence for the participation of the p.Val10Ile variant in non-pathogenic HBM phenotype and also provide regulatory evidence for certain enhancer elements in the *SOST* genomic region. In the future, more experiments are needed to gain a functional understanding of the regulation of sclerostin expression.

## 4. Materials and Methods

### 4.1. Cell Culture

The human osteosarcoma cell line Saos-2 was used for luciferase reporter assays, western blot experiments and 4C-seq assays. It was obtained from the American Type Culture Collection (ATCC^®^, Manassas, VA, USA) and grown in Dulbecco’s Modified Eagle Medium (DMEM; Sigma-Aldrich, St Louis, MO, USA), supplemented with 10% Fetal Bovine Serum (FBS; Gibco, Life Technologies, Carlsbad, CA, USA) and 1% penicillin/streptomycin (p/s; Gibco, Life Technologies, Carlsbad, CA, USA), at 37 °C and 5% of CO_2_. Human fetal osteoblast 1.19 (hFOB) cell line and human medulla-derived mesenchymal stem cells (MSCs) were used for 4C-seq assays. The hFOB 1.19 cell line was obtained from ATCC (ATCC^®^, Manassas, VA, USA) and grown in DMEM: F12 (1:1) medium without phenol red (Gibco, Life Technologies, Carlsbad, CA, USA), supplemented with 10% FBS and 0.3 mg/mL Geneticin (Gibco, Life Technologies, Carlsbad, CA, USA), at 34 °C and 5% of CO_2_. MSCs were kindly provided by Dr. José Manuel Quesada Gómez, from Instituto Maimónides de Investigación Biomédica, Hospital Universitario Reina Sofía, Córdoba, Spain [56]. These cells were grown in alpha-MEM medium (Gibco, Life Technologies, Carlsbad, CA, USA), supplemented with 10% FBS, 1% p/s and 1x Glutamax (Gibco, Life Technologies, Carlsbad, CA, USA), at 37 °C and 5% of CO_2_.

### 4.2. Luciferase Reporter Constructs, SOST Expression Vectors and Site-Directed Mutagenesis

The 520-bp fragment of the *SOST* upstream region (chr17:41,836,117-41,836,636, GRCh37/hg19) was PCR-amplified and cloned into the pGL3 basic Luciferase Reporter Vector (Promega, Madison, WI, USA). Constructs harboring the two alleles of the rare variant rs570754792 were cloned using the KpnI and BglII restriction sites (Primers used: ATACTAGGTACCGTCAAACAGAAACGCCTT and TATCTTAAGATCTCTTCCAGTAGCACAGGC). The SOST/pcDNA3.1+ expression vector was a gift from Xi He (Addgene plasmid # 10842; http://n2t.net/addgene: 10842; RRID: Addgene_10842). The missense mutations p.Val10Ile, p.Arg19His, p.Val21Leu (positive control), p.Ala28Val, p.Asn53del, p.Pro111Gln, p.Gly150Arg and p.Cys167Arg (positive control) were introduced into the expression vector using the QuickChange Site-Directed Mutagenesis kit (Agilent, Santa Clara, CA, USA), following the manufacturer’s instructions. In all cases, the presence of point mutations and absence of errors were verified through Sanger sequencing.

### 4.3. Luciferase Gene Reporter Assay

For the luciferase gene reporter assay, 3 × 10^5^ Saos-2 cells per well were cultured in 6-well plates, 24 h before the transfection. We transfected the cells with the renilla luciferase vector (pRL-TK) and either the empty pGL3 basic vector (negative control), the SOST-upstream region with each of the two alleles, or the basal promoter of *COL1A1* (positive control from [48]). The total amount of DNA transfected per well was 2.2 µg. Fugene HD was used following the manufacturer’s instructions. Forty-eight hours after transfection, cells were lysed and the luciferase activities of *Photinus pyrali* and *Renilla reniformis* were measured using a Glomax Multi+ luminometer (Promega, Madison, WI, USA), following the instructions of the Dual-luciferase reporter assay system (Promega, Madison, WI, USA). Each experiment included 3 replicates and was repeated independently in 4 separate experiments. The ratio of firefly and renilla luciferase measurements was calculated (relative luciferase units; RLU). All luciferase data were normalized using the empty vector activity (pGL3-basic).

### 4.4. Western Blot Assay

For the western blot assay, 3 × 10^5^ Saos-2 cells per well were cultured in 6-well plates, 24 h before transfection. We transfected 2 µg of the SOST expression plasmid (WT or any of the missense variants: p.Val10Ile, p.Arg19His, p.Val21Leu, p.Ala28Val, p.Asn53del, p.Pro111Gln, p.Gly150Arg and p.Cys167Arg). For the negative control, we transfected 2 µg of the empty pcDNA3 vector (data not shown). Fugene HD was used following the manufacturer’s instructions. After 24 h, the medium was changed, reducing from 2 to 1 mL of DMEM, without FBS or antibiotics. Forty-eight hours after transfection, the supernatant (conditioned medium) of each condition was collected and the cells were lysed with RIPA buffer. Proteins in the conditioned medium were concentrated using 10K Amicon Ultra filters (Merck-Millipore, Burlington, MA, USA). Extracellular proteins (concentrated conditioned medium) and intracellular proteins (RIPA extraction) were quantified using the PierceTM BCA Protein Assay Kit (ThermoFisher Scientific, Waltham, MA, USA). Proteins from concentrated conditioned medium (10 µg/lane) and intracellular proteins (5 µg/lane) were separated by electrophoresis in an SDS-polyacrylamide gel (SDS-PAGE) and transferred to a hydrophobic PVDF transfer membrane (Merck-Millipore, Burlington, MA, USA). We used the ab63097 antibody against sclerostin (Abcam, Cambridge, UK) and antibody against Tubulin (Abcam, Cambridge, UK) was used as the intracellular loading control. The images were developed using a peroxidase-conjugated secondary antibody [Anti-Rabbit IgG (A0545) for sclerostin antibody and Anti-Mouse IgG (A5906) for α-Tubulin antibody (Sigma-Aldrich, St Louis, MO, USA)]. For the extracellular protein, samples were obtained from 3 biological replicates that were quantified twice by BCA (ThermoFisher Scientific, Waltham, MA, USA). Each one of these quantifications was used to perform an SDS-PAGE experiment, producing a total of 2 experiments for each biological replica. For the intracellular protein, samples were obtained from 3 biological replicates and quantified once by BCA.

### 4.5. PCR Amplification and Sequencing

Resequencing of *SOST* and the *SOST* enhancer *ECR5* was performed in a small cohort of 11 HBM women described elsewhere [50]. Specifically, we resequenced the complete coding region, the 5′ and 3′ UTR regions and the intronic flanking regions of *SOST*, in addition to 1 Kb of the *SOST* proximal promoter and the *ECR5* enhancer region. PCR-amplification of each of these fragments was performed using the GoTaq Flexi DNA polymerase (Promega, Madison, WI, USA). The PCR fragments were analyzed by agarose gel electrophoresis and purification was done in MultiScreen TM Vacuum Manifold 96-well plates (Merck Millipore, Burlington, MA, USA). The purified PCR products were sequenced by the Sanger method in the CCiTUB genomic service (Genómica, Parc Cientific, Barcelona, Spain), using the BigDye™ Terminator v3.1 Cycle Sequencing Kit. Detection and electrophoresis were performed on automated capillary sequencer models 3730 Genetic Analyzer and 3730xl Genetic Analyzer. The sequence of the primers and the conditions used were previously described [47].

### 4.6. Bioinformatic Analysis and in Silico Predictions of the Effect of the Variants

To identify and characterize all the variants, we used information from the Ensembl database GRCh37.p13 (http://www.ensembl.org). Information was sought on the MAF of each of the variants found in the European population (EUR). We used SIFT (http://sift.bii.a-star.edu.sg/), PolyPhen (http://genetics.bwh.harvard.edu/pph2/), CADD (https://cadd.gs.washington.edu/) and REVEL [57] to test the effect of the missense variants. The SNP Function Prediction (snpinfo.niehs.nih.gov/snpinfo/snpfunc.html), MirSNP http://bioinfo.bjmu.edu.cn/mirsnp/), PolymiRTS Database 3.0 (http://compbio.uthsc.edu/miRSNP/search.php) and MicroSNiPer (http://vm24141.virt.gwdg.de/services/microsniper/) were used to analyse the binding of miRNAs to the variants in the 3′UTR region. Only those results that coincided with at least two different programs have been considered. The GTEx database (www.gtexportal.org/home/) was used to identify variants that act as eQTLs. The SNP Function Prediction (snpinfo.niehs.nih.gov/snpinfo/snpfunc.html) was used to predict transcription factor binding sites (TFBS) and new exonic splicing enhancers (ESE). To characterize regions that include the variants found, the UCSC Genome Browser GRCh37/hg19 was enriched with ENCODE data from osteoblasts.

### 4.7. 4C-Seq

4C-seq was carried out at the Functional Genomics Service of the Centro Andaluz de Biología del Desarrollo (CABD, Sevilla, Spain). 4C-seq libraries were generated from Saos-2, hFOB 1.19 and hMSC cell lines as described previously [58,59]. Four-bp cutters were used as primary (DpnII) and secondary (Csp6I) restriction enzymes. For each cell line, a total of 1.6 µg of library was PCR amplified (primers used: CCTGCTGTGACATCTTCTGC and CCATTCAGAGGGGTGTGGATC). Samples were sequenced with Illumina Hi-Seq technology according to standard protocols at the Genomics Service of the Centro Nacional de Investigaciones Cardiovasculares (CNIC, Madrid, Spain). 4C-seq data were generated as described previously [60]. Briefly, raw sequencing data were demultiplexed and mapped to the corresponding reference genome (GRCh37). Reads located in fragments flanked by two restriction sites of the same enzyme, in fragments smaller than 40 bp or within a window of 10 kb around the viewpoint were filtered out. 4C-seq data were normalized by the total weight of reads within ±2 Mb around the viewpoint.

4C-seq data were analyzed with the AFourC software (Martínez-Gil, N., Herrera, C., Gritti, N., Roca-Ayats, N., Ugartondo, N., Garcia-Giralt, N., Ovejero, D., Nogués, X., Garcia-Fernandez, J., Grinberg, D. & Balcells, S., submitted) following and adapting previously described pipelines [61,62]. Briefly, we assumed that the 4C signal profile relative to the viewpoint *v* with coordinate *xv* on chromosome *N* is formed by three independent contributions: a constant background level, a negative exponential representing the monotonic decay of the 4C signal with the genomic distance from the viewpoint [62] and *N* gaussians representing the interaction peaks:S4Cx=B+Ixv·e−x−xv/λ+∑i=1NPxi·e−x−xi22·σ2

To estimate the genomic distance-dependent monotonic decay, we assumed a symmetric trend around the viewpoint and performed the exponential fit on the left-right averaged profile. Statistically significant peaks were detected using a *p*-value of 0.0005. The software package is publicly available at https://github.com/Nikoula86/AFourC.git.

### 4.8. Statistical Analyses

In the luciferase assays, the differences between the two alleles were evaluated using the Welch Two Sample t-test, and the homogeneity of variances was confirmed using Bartlett test and normality was confirmed using Shapiro test. Statistical significance was set at *p*-value < 0.05. The statistical evaluation was carried out with Rstudio Version 3.6.1. The representations were performed using Graphpad (Graphpad Prism 8, San Diego, CA, USA).

In the analysis of the western blots, we used the ImageJ 1.52a software to quantify the intensity of the bands and the GraphPad 8 software for the graphical representation of the means and standard deviations of each mutant sclerostin.

The specific statistics for the 4C-seq experiments have been detailed in Section 4.7.

## Figures and Tables

**Figure 1 ijms-22-00489-f001:**
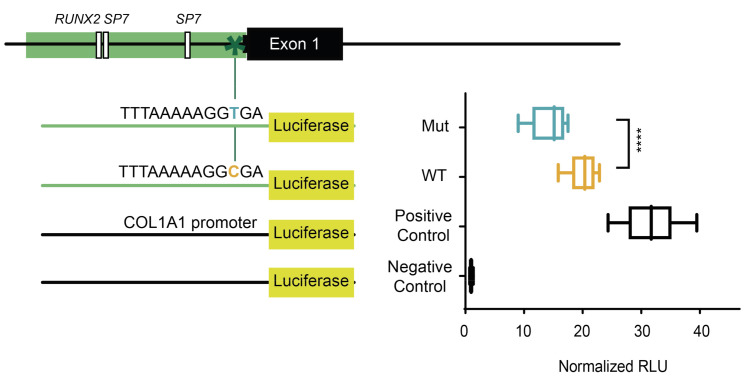
Functional testing of a rare variant in the extended TATA box of *SOST*. rs570754792 (g.41836179 G > A in GRCh37.p13) belongs to the extended TATA box of *SOST*, which is encoded in the negative strand of chromosome 17, thereby generating a C > T substitution. In black, a scheme of the location of the variant (asterisk) in the *SOST* upstream region and in green the constructs tested. We used the luciferase gene without promoter as a negative control and the *COL1A1* promoter as a positive control. Relative luciferase units (RLU; see material & methods) are normalized to the negative control. The experiment was performed four times in triplicate. **** indicates *p* < 0.0001.

**Figure 2 ijms-22-00489-f002:**
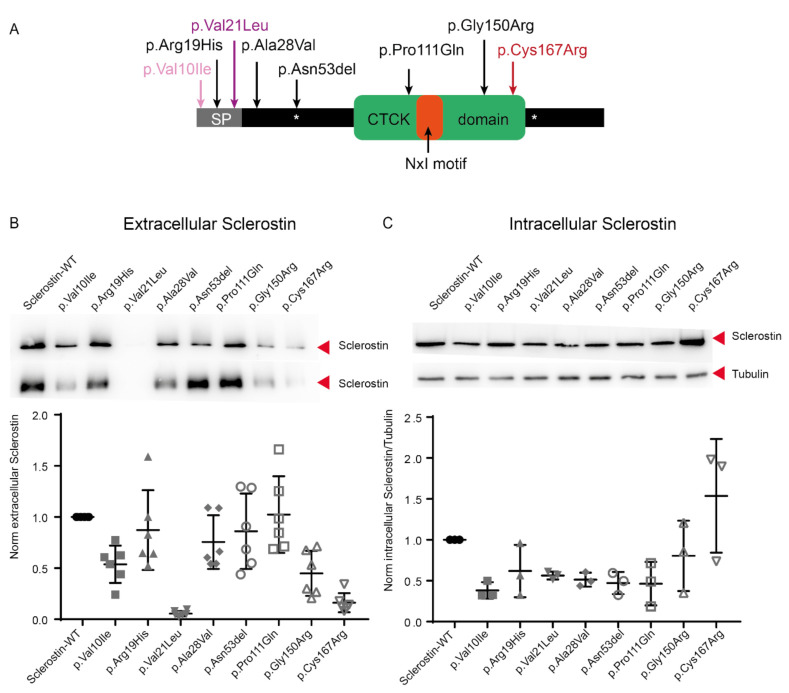
Western blot of variant sclerostin proteins transfected into Saos-2 cells. Variants were generated by in vitro mutagenesis into a *SOST* expression vector and subsequently transfected and heterologously expressed in Saos-2 cells. (**A**) Tested amino acid variants, displayed along a scheme of the sclerostin, with indication of specific domains (SP: signal peptide; CTCK: *C*-terminal cystine-knot; NXI:NXI(R/K) sequence motif; asterisks denote the glycosylation *N*-linked (GlcNAc) asparagine). In red, the mutation present in a patient with sclerosteosis [17]; in purple, the mutation found in a patient with CDD [10] and in pale pink, the mutation associated with high BMD [47] (**B**) Detection of sclerostin in the extracellular media. Above, image of the first biological replicate; below, quantification of the sclerostin bands from images of 6 equivalent blots, normalizing to the extracellular levels of WT sclerostin. (**C**) Detection of intracellular sclerostin. Above, image of the first biological replica of sclerostin and their corresponding tubulin; below, the quantification of the ratios of sclerostin/tubulin protein from images of 3 equivalent blots, normalizing to the ratio of WT sclerostin/tubulin.

**Figure 3 ijms-22-00489-f003:**
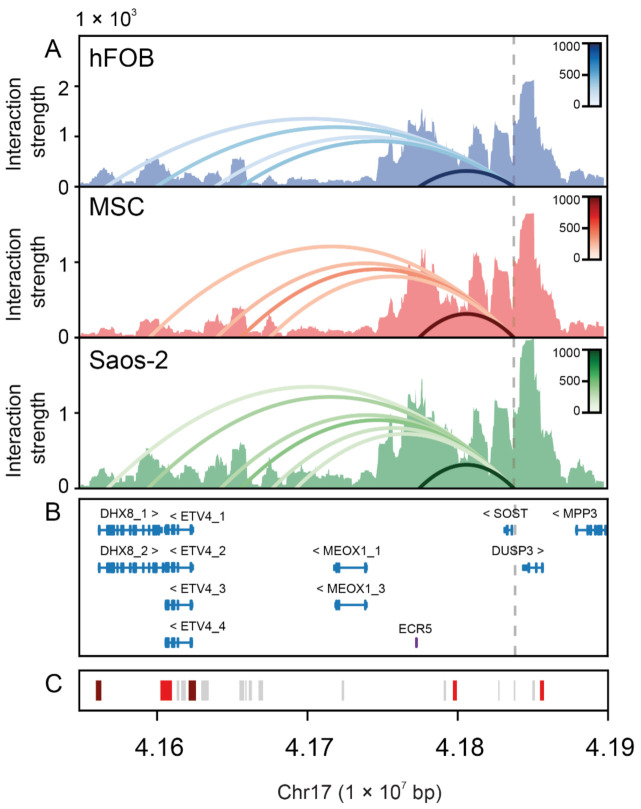
Chromatin interactions of the *SOST* proximal promoter region as detected by 4C-seq in human fetal osteoblasts (hFOB), mesenchymal stem cells (MSC) and the Saos-2 cell line. Panel (**A**): Graphical representation of a 350-kb Chr.17 region containing *SOST* and all the genomic elements found to interact with it displaying both the read depths (Interaction strength) and their corresponding significance (arched lines). The dashed vertical line marks the *SOST* promoter, used as the viewpoint. Color scale corresponds to the strength of the significant interactions. Panel (**B**): Genes in this region: DHX8_1 (ENST00000262415.3), DHX8_2 (ENST00000540306.1), ETV4_1 (ENST00000319349.5), ETV4_2 (ENST00000545954.1), ETV4_3 (ENST00000591713.1), ETV4_4 (ENST00000545089.1), MEOX1_1 (ENST00000520305.1), MEOX1_3 (ENST00000393661.2), SOST (ENST00000301691.2), DUSP3 (ENST00000226004.3), MPP3 (ENST00000398389.4) from GRCh37/hg19. In purple the ECR5 position [35]. Panel (**C**): Representation of the GeneHancer (GH) track (enhancers -grey- and promoters -red-).

**Table 1 ijms-22-00489-t001:** Variants in *SOST* and *ECR5* found in women with an HBM phenotype.

Position	rs Number	Type	MAF (EUR)	MAF (HBM Cohort)	Predicted Functionality
*SOST*					
g.41838229C > T	rs1237278	5′ UP	0.355 (C)	0.45 (C)	eQTL; TFBS
g.41837719G > A	rs851058	5′ UP	0.402 (A)	0.35 (A)	eQTL; TFBS
g.41837510_41837512del	rs10534024	5′ UP	0.352 (TCCTCCT)	0.45 (TCCTCCT)	eQTL
p.Val10Ile	rs17882143	M	0.018 (T)	0.05 (T)	eQTL T; B; LB
c.-22G > A	rs17885799	5′ UTR	-	0.05 (A)	TFBS, ESE
c.*320C > T	rs17883310	3′ UTR	0.013 (T)	0.05 (T)	miRNA
c.*1004G > A	rs17886183	3′ UTR	0.004 (A)	0.05 (A)	eQTL miRNA
c.*1266_*1267insG	rs17881550	3′ UTR	0.412 (ins)	0.35 (ins)	eQTL
*ECR5*					
g.41774092G > A	rs552004150	3′ D	0.003 (A)	0.05 (A)	

M: Missense variant; 5′ UP: 5′ Upstream; 5′ UTR: 5′ UTR variant; 3′ UTR: 3′ UTR variant; 3′ D 3′ downstream variant; MAF(EUR) from 1000 genomes. eQTL: described as eQTL in GTEx. TFBS: Transcription factor binding site; ESE: Exon Splicing enhancer; T: Tolerated by SIFT; B: Benign by Polyphen-2; LB: Likely Benign by CADD; miRNA: allelic differences in the miRNA binding prediction.

## Data Availability

Data is contained within the article or supplementary material and the software for 4C-seq analysis is publicly available at https://github.com/Nikoula86/AFourC.git.

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
