# Peer review of "Genetics and Genomics of SOST: Functional Analysis of Variants and Genomic Regulation in Osteoblasts"

_ijms, 2021, doi:10.3390/ijms22020489_

Round 1
Reviewer 1 Report
The MS "Genetics and genomics of SOST: functional analysis of variants and genomic regulation in bone cells" by Núria Martínez-Gil et al has been described and will advance the current scientific knowledge. However, the authors need to address the following issues to further proceed.
The Renilla luciferase activity may be used to normalize the values instead of control.
.... Sost levels in osteocytes [39]. I think this should be a Sclerostin level.
"Taking into account the crucial role of SOST in bone metabolism,....." Is this supposed to be bone growth?
It would be nice to include SNP data from populations regarding rs570754792 and rs17882143?
Is it possible to measure promoter activity in cells without any treatment?
Why there are two WB for extracellular sclerostin but one for intracellular sclerostin?
Statistical analysis has been mentioned only for luciferase assays, but not other graphs. A separate section may be used for overall statistical analysis.
Author Response
The MS "Genetics and genomics of SOST: functional analysis of variants and genomic regulation in bone cells" by Núria Martínez-Gil et al has been described and will advance the current scientific knowledge. However, the authors need to address the following issues to further proceed.
The Renilla luciferase activity may be used to normalize the values instead of control.
Indeed, this is actually what we have done: for each transfection, both firefly and renilla luciferase constructs are mixed, and the firefly luciferase readout is divided by the renilla readout to “normalize” it (as explained in heading 4.3 Luciferase gene reporter assay, page 9 of 15). Thereafter, when comparing the experimental promoter fragments, we like to use an empty vector (luciferase luciferase without a promoter) as reference, arbitrarily set at 1.
.... Sost levels in osteocytes [39]. I think this should be a Sclerostin level.
We thank the reviewer for the comment. We have changed it in the text of the manuscript (page 2 of 15, highlighted in yellow).
"Taking into account the crucial role of SOST in bone metabolism,....." Is this supposed to be bone growth?
Yes, we have changed the word metabolism to growth to make it more specific (page 2 of 15, highlighted in yellow).
It would be nice to include SNP data from populations regarding rs570754792 and rs17882143?
We agree with the reviewer that this may add useful information to the reader. We have now included the frequency of the minor allele of these two variants according to gnomAD (pages 2 and 3 of 15, highlighted in yellow).
Is it possible to measure promoter activity in cells without any treatment?
Yes, it is possible to measure the promoter activity in the cells by RT-qPCR of SOST transcripts. However, the variant is rare, and it would be very difficult to find homozygotes of whom we would need an osteoblast sample. Unfortunately, we are left with the option of transfecting plasmid.
Why there are two WB for extracellular sclerostin but one for intracellular sclerostin?
We performed two WBs for extracellular sclerostin because we did not have an adequate control of loading. That is why the proteins of each replica were quantified twice separately and two independent WBs were performed to minimize loading errors that cannot be minimized by loading control. On the other hand, in intracellular sclerostin we had a good loading control that allowed us to correct these small differences in load.
Statistical analysis has been mentioned only for luciferase assays, but not other graphs. A separate section may be used for overall statistical analysis.
Following the reviewer’s suggestion, we have added a final subheading in M&M entitled 4.8. Statistical analyses (page 11 of 15, highlighted in yellow).

Reviewer 2 Report
Nice idea. I was wondering if the analysis could be performed also on other osteoblast cell lines, such as 143b that are more aggressiv and metastatic.
Interestingly could be also the functional analysis of these variants in osteoblast cells and evaluate how these variants can change markers (ALP, gene expression) or activity of osteoblast cells; but probably this could be another study.
I suggest to modify the title, no bone cells, you describe only osteoblast, so, probably bone cells is not properly correct.
Author Response
Nice idea. I was wondering if the analysis could be performed also on other osteoblast cell lines, such as 143b that are more aggressiv and metastatic. Interestingly could be also the functional analysis of these variants in osteoblast cells and evaluate how these variants can change markers (ALP, gene expression) or activity of osteoblast cells; but probably this could be another study.
We thank the reviewer for the kind words. Indeed, it would be interesting to try other osteoblastic cell lines or even some more osteocyte-like lines. Likewise, measuring markers of osteoblast activity would add new insights. We will consider these suggestions for future work.
I suggest to modify the title, no bone cells, you describe only osteoblast, so, probably bone cells is not properly correct.
Following this reviewer’s suggestion, we have changed “bone cells” to “osteoblasts” in the title (highlighted in yellow).
